# Rapid Destructive Arthrosis Due to Subchondral Insufficiency Fracture of the Shoulder: Clinical Characteristics, Radiographic Appearances, and Outcomes of Treatment

**DOI:** 10.3390/diagnostics10110885

**Published:** 2020-10-30

**Authors:** Chul-Hyun Cho, Byung-Woo Min, Kyung-Jae Lee, Jun-Young Kim, Du-Han Kim

**Affiliations:** 1Department of Orthopedic Surgery, Keimyung University Dongsan Hospital, Keimyung University School of Medicine, Daegu 4260, Korea; oscho5362@dsmc.or.kr (C.-H.C.); min@dsmc.or.kr (B.-W.M.); oslee@dsmc.or.kr (K.-J.L.); 2Department of Orthopedic Surgery, School of Medicine, Catholic University, Daegu 42601, Korea; dr.junyoung@gmail.com

**Keywords:** rapid destructive arthrosis, subchondral fracture, shoulder joint humeral head replacement, osteoarthritis, osteoporosis, shoulder arthroplasty

## Abstract

The purpose of our study was the clinical characteristics, radiographic appearance, and outcomes after treatment in patients with rapid destructive arthrosis (RDA) due to subchondral insufficiency fracture (SIF) of the shoulder. Twenty-two cases of RDA of the shoulder were retrospectively reviewed. Clinical outcomes for 15 cases who underwent shoulder arthroplasty were evaluated at an average of 41.4 months. The mean age of patients was 73.7 years (range 50–83 years), and there were 20 women and 2 men. The mean time from onset of symptoms to head collapse was 6.8 months (range 1–12 months). The mean t-score of bone mineral density was −3.1. Nine patients had pseudoparalysis. Based on radiographic appearance, a diversity of types of head destruction with subchondral fracture, bone marrow edema, joint effusion, and synovitis were observed in all cases. In conclusion, RDA due to SIF of the shoulder, presenting with severe short-term pain and functional disability, commonly occurred in elderly women with bone fragility. MRI revealed bone marrow edema, extensive joint effusion, and synovitis as well as a diversity of types of head destruction with subchondral fracture within several months from onset of symptoms.

## 1. Introduction

Rapid destructive arthrosis (RDA) is the marked destruction of a joint within months after the onset of symptoms [1]. This condition usually occurs in the hip and more rarely in the shoulder [2]. Nguyen [3] reported that RDA of the shoulder mainly occurs in elderly women and is characterized by bone destruction, joint effusion, basic calcium phosphate crystals, and rotator cuff tears. However, its etiology and pathogenesis is poorly understood. Because of the unclear pathogenesis and clinical characteristics of RDA of the shoulder, the diagnosis and treatment of this condition are difficult for many physicians.

Very recently, subchondral insufficiency fractures (SIFs) of the shoulder have been described as a cause of rapid joint destruction [1,2,4,5]. SIF is a recently proposed concept and is thought to cause femoral head collapse associated with RDA of the hip [6,7]. It may be related to repetitive microtrauma and occurs immediately below the articular cartilage of a weight-bearing joint [8]. It was thought to occur secondary to physiological stress applied to a weakened bone due to reduced bone mineralization, commonly seen in elderly women with osteoporosis [9]. Although the pathogenesis of RDA of the shoulder remains unclear, there have been several reports for RDA resulting from SIF of the shoulder [1,2,4,5]. Magnetic resonance imaging (MRI) plays an important role in the diagnosis of SIF, which is often inconspicuous on initial radiographs [6]. MRI findings included an irregular serpiginous low-intensity band convex to the articular surface on T1-weighted images, as a suggestive finding of subchondral fracture, large amount of joint effusion, and bone marrow edema in the humeral head and metaphyseal area [6].

To the best of our knowledge, only 25 cases of RDA due to SIF of the shoulder have been reported as case reports or small case series [1,2,4,5,10,11]. Because of its rarity and related gaps in knowledge, this condition might be misdiagnosed as idiopathic destructive arthritis of the shoulder, osteonecrosis of the humeral head, Milwaukee shoulder syndrome, infectious arthritis, or osteomyelitis [12,13]. Most studies reported that SIF of the humeral head occurs in elderly women with bone fragility [1,2,4,5,11]. Considering the recent increase in life expectancy, we believe that RDA due to SIF of the shoulder may increase with an increased understanding of the clinical features and radiographic findings supporting its diagnosis.

The primary aim of this study was to investigate the clinical characteristics and radiographic appearance in patients with RDA due to SIF of the shoulder. The secondary aim was to evaluate clinical outcomes after shoulder arthroplasty in these patients.

## 2. Materials and Methods

This study was approved by the institutional review board (IRB No: 202003042), approved on 19 March 2020. This retrospective study included 311 patients who underwent or scheduled shoulder arthroplasty at a single institution between 2012 and 2019. Inclusion criteria were as follows: (1) patients with RDA of the shoulder joint; rapid destruction was defined as >25% erosive osteolysis of the humeral head within 6 months according to Lequesne et al. [14] or >2 mm (or 50%) joint space narrowing in 12 months according to Postel and Kerboull [15]; (2) available medical records and radiographic findings; (3) SIF of the humeral head confirmed by a shoulder specialist and musculoskeletal radiologist. Exclusion criteria included: (1) a history of trauma; (2) infectious arthritis; (3) inflammatory arthritis; (4) osteonecrosis of the humeral head; (5) neuropathic arthropathy; or (6) crystal-induced destructive arthropathy. Among 311 patients, 289 patients were excluded including 215 cuff tear arthropathy or massive rotator cuff tear, 50 osteoarthritis, 16 infectious arthritis, and 8 osteonecrosis of the humeral head. Finally, 22 patients were included in this study.

### 2.1. Demographic and Clinical Evaluation

All available demographic and clinical evaluations were assessed, including age, sex, side, body mass index (BMI), occupation, bone mineral density (BMD), history of trauma, history of oral steroid therapy, history of alcohol abuse, duration of symptoms, clinical scores (i.e., visual analog scale (VAS) pain score, American Shoulder and Elbow Surgeons (ASES) score, subjective shoulder value (SSV)), and range of motions (ROMs) (forward flexion, abduction, external rotation with the arm at the side, and internal rotation at the back).

### 2.2. Radiographic Examination

To diagnose RDA due to SIF, plain radiographs, computed tomography (CT), and MRI were performed. MRI was performed using a 1.5 Tesler scanner (Avanto; Siemens, Erlangen, Germany) with a dedicated shoulder coil. We obtained MR T1/T2-weighted coronal, sagittal, and axial images and T2-weighted fat suppression images in at least one plane. One patient did not undergo MRI because of pacemaker insertion and received ultrasonography instead.

Based on plain radiographs, CT, and MRI, we analyzed humeral head involvement, humeral head destruction, bone marrow edema, joint effusion, bone debris and calcification, synovitis, rotator cuff tear, arthritic change, and glenoid involvement. Humeral head involvement was classified as follows based on involvement of the entire surface of the humeral head: grade 1 (1/3 involvement), grade 2 (1/3–2/3 involvement), and grade 3 (>2/3 involvement). Humeral head destruction was classified as follows based on destruction of the height of the humeral head: grade 1 (<1/3 destruction), grade 2 (1/3–2/3 destruction), and grade 3 (>2/3 destruction).

### 2.3. Laboratory Tests and Histologic Examinations

To rule out infectious arthritis, inflammatory arthritis, or crystal-induced arthropathy, blood tests including differential white blood cell (WBC) count, erythrocyte sedimentation rate (ESR), C-reactive protein (CRP) level, and fluid analysis of joint aspiration were checked. Histologic examination for resected specimens of humeral head or soft tissues was also performed.

### 2.4. Treatment

Twenty out of 22 patients underwent operative treatment. Two patients who scheduled shoulder arthroplasty decided on conservative treatment instead because of severe restriction of cardiac function or poor general condition caused by several medical diseases. Sixteen patients underwent primary shoulder arthroplasty including 14 reverse total shoulder arthroplasty (RTSA), 1 anatomic total shoulder arthroplasty (TSA), and 1 hemiarthroplasty (HA). Three patients underwent 2-stage RTSA after open debridement and cement spacer insertion because infectious arthritis or osteomyelitis could not be completely ruled out. Two patients underwent secondary RTSA after failed arthroscopic debridement (Figure 1). A young, active patient with grade 1 destruction had arthroscopic debridement.

### 2.5. Statistical Analysis

The IBM SPSS ver. 22.0 (IBM Co., Armonk, NY, USA) was used for statistical analysis. To compare the preoperative and final clinical scores and ROMs, we used the Wilcoxon signed rank test. Statistical significance was set at *p* < 0.05.

## 3. Results

The mean age of the patients was 73.7 ± 8.1 years (range 50–83 years), and there were 20 women and two men. The right shoulder was involved in 18 patients and the left shoulder in four. The mean BMI was 24.7 ± 3.1 kg/m^2^ (range 17.8–29.8 kg/m^2^), and the mean t-score of BMD tests was −3.1 ± 1.0 (range −5.2 to −2.1) with osteoporosis in 14 patients and osteopenia in five. Three patients did not perform BMD test. No patients had a history of trauma, oral steroid therapy, or alcohol abuse (Table 1).

The mean time from onset of symptoms to head destruction was 6.8 months (range 1–12 months). MRI revealed subchondral fractures that presented with a low signal intensity band on T1-weighted images and high signal intensity on T2-weighted images. A diversity of types of head destruction with subchondral fracture, bone marrow edema, joint effusion, and synovitis were observed in all cases. Bone debris and calcification was observed in 14 patients (63.6%) and rotator cuff tears were observed in 17 cases (77.3%). Arthritic change in the glenohumeral joint was observed in five patients (22.7%) (Table 2 and Figure 2). Glenoid involvement was observed in three patients (13.6%). Humeral head involvement was classified as grade 3 (*n* = 13 shoulders), grade 2 (*n* = 8 shoulders), and grade 1 (*n* = 1 shoulder). Humeral head destruction was classified as grade 1 (*n* = 11 shoulders), grade 3 (*n* = 7 shoulders), and grade 2 (*n* = 4 shoulders) (Figure 3).

The mean WBC count, neutrophil count, ESR, and CRP were 7395/µL (normal range 4000–10,000/µL), 66% (normal range 38–73%), 31.1 mm/h (normal range 0–30 mm/h), and 0.72 mg/L (normal range 0–0.5 mg/L), respectively. No growth of pathogens or crystals was observed in aspirated joint fluid analysis and intraoperative tissue culture.

Intraoperatively, resected humeral heads were collapsed with detached articular cartilage. In cases of severe head destruction, articular surfaces were flat and covered by fibrous tissues. Histologically, slender bone trabeculae were identified, and bone marrow was replaced by fat tissue. There was evidence of partial osteoid formation admixed with multinucleated osteoclasts in the fibrotic background. Palisading osteoclasts around the newly formed bone were noted. No significant inflammatory infiltrate, infarcts, and osteonecrosis were identified (Figure 4).

Fifteen patients had follow-up periods longer than 12 months after shoulder arthroplasty. The mean follow-up period was 41.4 months (range 12–117 months). The mean VAS pain score, ASES score, and SSV improved, respectively, from 7.4, 25.0, and 26.7% preoperatively to 1.6, 79.8, and 76% postoperatively (*p* < 0.001). The mean active forward flexion, abduction, external rotation at side, and internal rotation at the back improved, respectively, from 66.3°, 57.0°, 22.3°, and the 5th lumbar vertebral level preoperatively to 140.3°, 122.0°, 54.0°, and the 2th lumbar vertebral level postoperatively (*p* < 0.001). Two complications following shoulder arthroplasty occurred, fixation failure (*n* = 1) and loosening of glenoid (*n* = 1); both patients were treated by RTSA with glenoid bone graft for preoperative glenoid defects. A patient with glenoid fixation failure underwent resection arthroplasty with poor clinical outcome. A patient with glenoid loosening was observed with serial evaluations.

## 4. Discussion

The present study revealed RDA due to SIF of the shoulder, presenting with short-term severe pain and functional disability, commonly occurred in elderly patients with bone fragility. MRI revealed bone marrow edema, extensive joint effusion, and synovitis as well as a diversity of types of head destruction with subchondral fracture. The results presented here indicate that SIF should be included in the differential diagnosis of acute onset shoulder pain in elderly patients.

Since Tokuya et al. [2] in 2004 first reported a case with SIF of the shoulder resulting in RDA, only 25 cases have been reported as case reports or small case series (Table 3) [1,2,4,5,10,11]. In a review of the literature, most RDAs due to SIF of the shoulder occurred in women older than 70 years with bone fragility. Kekatpure et al. [10] did not report whether or not BMD tests were performed. All reported cases had osteoporosis or osteopenia except the study by Kekatpure et al. [10]. The present study revealed that the mean age of the patients was 73.7 years, and mean t-score of BMD tests was −3.1. All patients were diagnosed with osteoporosis (*n* = 14) or osteopenia (*n* = 5), except for the three patients who did not receive a BMD test. It is worth noting that our study included two men (Case 14, 81 years old; Case 20, 75 years old) with RDA due to SIF of the shoulder. In previous studies, all reported cases were in women. Taken together, RDA due to SIF of the shoulder commonly occurred in elderly women with bone fragility.

Yoshikawa et al. [5] reported the time from onset of symptoms to head destruction was 3 and 8 months in two cases. Meanwhile, Tokuya et al. [2] reported a case of RDA due to SIF of the humeral head and glenoid, in which head destruction occurred within 1 month after the first visit. Kekatpure et al. [10] reported that nine cases of RDA involved head destruction within an average 5.7 months after the initial symptoms. Kim et al. [11] described that flattening of the humeral head within an average 4.1 months after onset of symptoms is a key characteristic of RDA of the shoulder. In the present study, the mean time from onset of symptoms to head destruction was 6.8 months (range 1–12 months). Taken together, RDA caused by SIF of shoulder can be defined as collapse and flattening of the humeral head within 12 months after initial onset of symptoms.

MRI plays an important role in the diagnosis of SIF, which is often inconspicuous on initial radiographs [8]. MRI reveals an irregular serpiginous low-intensity band convex to the articular surface on T1-weighted images as a suggestive finding of SIF [6]. Additional MRI findings included extensive joint effusion and bone marrow edema in the humeral head and metaphyseal area with a diffuse low-intensity area on T1-weighted images and high intensity area on T2-weighted images [6]. Rotator cuff tear was accompanied in approximately 50% of previous reported cases. These findings reported here are consistent with those of previous studies. In the present study, a subchondral low-intensity band was confirmed on the T1-weighted image, and bone marrow edema was observed in all cases. Seventeen out of 22 cases (77.3%) had rotator cuff tears, among them, 12 involved more than two rotator cuff tendons.

Differential diagnoses of RDA due to SIF of the shoulder included osteonecrosis of the humeral head, rheumatoid arthritis, infection, crystal-induced arthritis, hemodialysis-related destructive arthropathy, osteoarthritis, and neuropathic arthropathy. Because SIF is a recently proposed disease entity, RDA due to SIF of the shoulder might be misdiagnosed as idiopathic destructive arthropathy, osteonecrosis of the humeral head, Milwaukee shoulder syndrome, infectious arthritis, or osteomyelitis [12,13]. One of the differential points with osteonecrosis of the humeral head is that osteonecrosis shows preserved articular cartilage and joint space until the advanced stage. Ikemura et al. [16] suggested that osteoporotic elderly women without any history of oral steroid therapy or alcohol abuse need to first be considered to have SIF when plain radiographs show a collapse of the femoral head. In the present study, we tried to rule out these conditions through the patients’ medical history, blood tests, synovial fluid analysis, imaging studies, intraoperative culture, and histologic examination. However, it was difficult to rule out infectious arthritis or osteomyelitis in three cases, because they had abnormal laboratory and radiographic findings (e.g., focal destruction of the humeral head, bone marrow edema, extensive joint effusion). Therefore, two-stage RTSA after open debridement and cement spacer insertion was used in these patients, although a final diagnosis of SIF was made based on radiographic, intraoperative, and histologic findings. Judging by our results, it is proposed that SIF should be included in the differential diagnosis of acute onset shoulder pain in elderly patients with bone fragility.

Previous studies reported that histologic findings of RDA due to SIF of the shoulder included fragmented bone trabeculae, osteoid formation around the newly formed bone, increased osteoclast, and fracture callus [1,2,4,5,10,11]. The results of the present study are consistent with the findings of these previous studies. However, histologic diagnosis of RDA could not be confirmed in cases with a completely destroyed head. Therefore, short-term severe shoulder pain and functional disability, radiographic findings, and histologic findings were key factors in the differential diagnosis of severe RDA cases.

Similar to RDA of the hip, RDA of the shoulder may not respond well to conservative treatment. In a review of the literature, all reported cases with RDA due to SIF of the shoulder underwent arthroplasty [17]. Our cases were treated by shoulder arthroplasty (*n* = 17) or arthroscopic debridement (*n* = 3) depending on the patient’s age, activity, symptoms, degree of head destruction, and presence or absence of a rotator cuff tear [18,19]. However, two cases with failed arthroscopic debridement underwent secondary RTSA. Therefore, the results from our study suggest that treatment of choice in patients with RDA due to SIF of the shoulder is shoulder arthroplasty. Kim et al. [11] highlighted that shoulder arthroplasty should be performed as early as possible, because glenoid destruction can be seen in rare cases of severe progression of head destruction in patients with RDA of the shoulder joint. The authors agree with this opinion. In the present study, shoulder arthroplasty for patients without glenoid defect yielded satisfactory clinical outcome. However, two complications (fixation failure (*n* = 1) and loosening of glenoid after RTSA (*n* = 1)) were observed in patients with glenoid defect. Therefore, shoulder arthroplasty for these conditions should be performed before the glenoid defect occurs in an attempt to avoid postoperative complications.

This study has a few limitations. First, it is a retrospective study. Second, it was difficult to understand the natural course of the disease, because it is not a longitudinal study. Third, we did not elucidate the pathogenesis of SIF of the shoulder as a non-weight-bearing joint. Further prospective studies are needed to elucidate pathogenesis, associated factors, and natural history of this disease. However, it is of note that this is the first study to investigate the clinical characteristics, radiologic appearances, treatments, and outcomes of a large series of RDA due to SIF of the shoulder.

In conclusion, RDA due to SIF of the shoulder, presenting with short-term severe pain and functional disability, commonly occurred in elderly women with bone fragility. MRI revealed bone marrow edema, extensive joint effusion, and synovitis as well as a diversity of types of head destruction with subchondral fracture within several months from onset of symptoms.

## Figures and Tables

**Figure 1 diagnostics-10-00885-f001:**
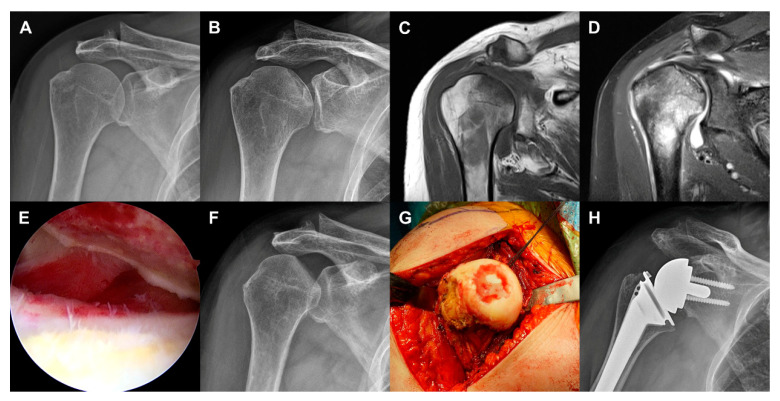
Case 10 (59-year-old woman). (**a**) No evidence of head collapse in plain radiograph at the time of onset of symptoms. (**b**) Subchondral fracture in plain radiograph taken at 12 months after onset of symptoms. (**c**) Low signal intensity band on T1-weighted MR image. (**d**) Bone marrow edema, joint effusion, and rotator cuff tear on T2-weighted MR image. (**e**) Cartilage defect and exposure of subchondral bone in arthroscopic view. (**f**) Grade 2 head involvement and grade 1 head destruction in plain radiograph taken before reverse total shoulder arthroplasty (RTSA). (**g**) Intraoperative photograph. (**h**) Plain radiograph after RTSA.

**Figure 2 diagnostics-10-00885-f002:**
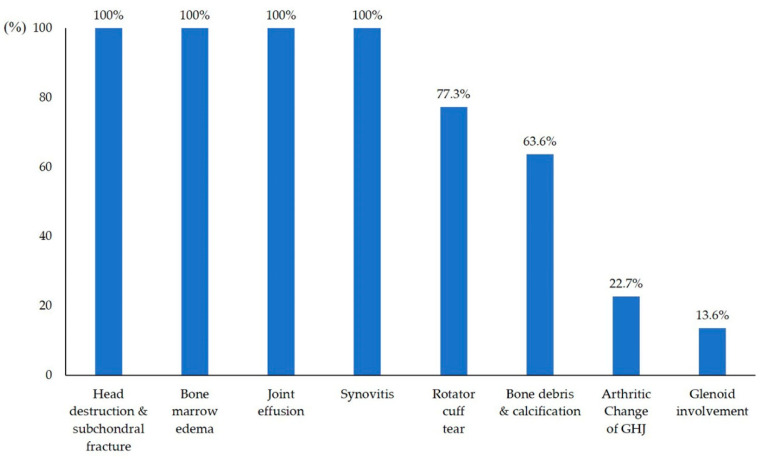
Incidence of radiographic findings.

**Figure 3 diagnostics-10-00885-f003:**
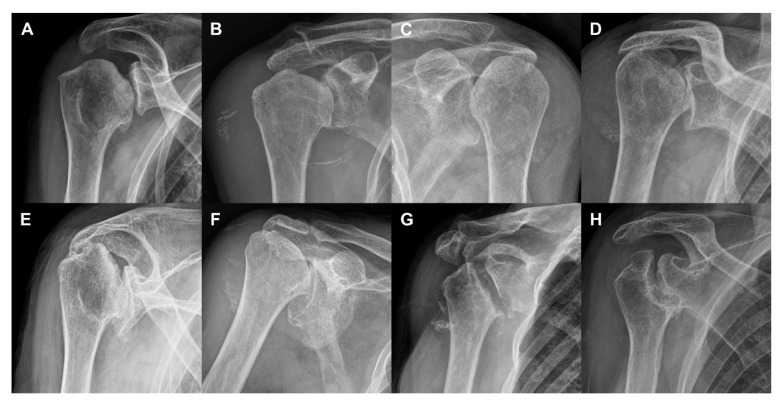
Diversity of humeral head involvement and destruction. (**a**) Case 8 (grade 1 involvement, grade 1 destruction). (**b**) Case 7 (grade 2, grade 1). (**c**) Case 11 (grade 2, grade 1). (**d**) Case 19 (grade 3, grade 3). (**e**) Case 16 (grade 3, grade 2). (**f**) Case 14 (grade 3, grade 3). (**g**) Case 21 (grade 3, grade 3). (**h**) Case 22 (grade 3, grade 3).

**Figure 4 diagnostics-10-00885-f004:**
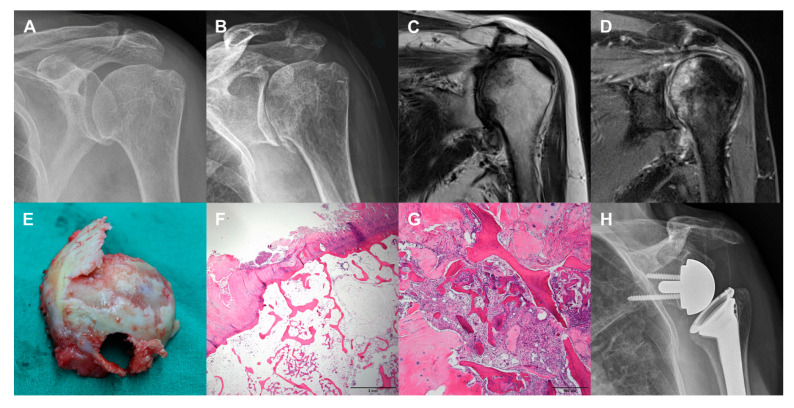
Case 1 (81-year-old woman). (**a**) No evidence of head collapse in plain radiograph at the time of onset of symptoms. (**b**) Subchondral fracture in plain radiograph taken at 4 months after onset of symptoms. (**c**) Low signal intensity band on T1-weighted MR image. (**d**) High signal intensity on T2-weighted MR image. (**e**) Exposure of subchondral bone with detached articular cartilage in photograph of resected humeral head. (**f**) Histologic findings showing degenerated and fibrillar articular surface and osteoid formation (hematoxylin-eosin stain, ×1.25). (**g**) Histologic findings showing regenerating bony trabeculae and fibrosis (hematoxylin-eosin stain, ×10). (**h**) Plain radiograph after RTSA.

**Table 1 diagnostics-10-00885-t001:** Patient clinical data.

Case	Age	Sex	Side	BMI	Occupation	BMD (t-Score)	Time from Sx Onset to Head Collapse (month)	VAS Pain Score	ASES Score	WBC Count (Neutrophil Count)	ESR	CRP
1	81	F	Lt	21.5	House wife	−3.0	4	10	3.3	8030 (67%)	35	0.20
2	75	F	Rt	26.6	Farmer	−2.4	12	10	3.0	6240 (58%)	48	0.02
3	68	F	Rt	26.7	Farmer	−2.1	10	7	31.7	4460 (51%)	27	0.10
4	83	F	Rt	21.1	House wife	−3.0	12	8	18.3	6040 (64%)	13	0.01
5	73	F	Rt	17.8	Farmer	−3.3	6	4	50.0	5620 (67%)	8	0.01
6	77	F	Rt	28.4	Farmer	−3.0	12	8	18.3	7980 (54%)	32	0.10
7	72	F	Rt	22.2	Farmer	−4.8	3	5	39.2	10,180 (70%)	33	0.24
8	81	F	Rt	24.6	House wife	−2.4	3	9	11.7	10,420 (80%)	22	0.01
9	64	F	Lt	26.0	Farmer	−2.4	12	6	45.0	5620 (56%)	8	0.03
10	59	F	Rt	27.2	Restaurant work	−3.0	12	6	23.3	7670 (61%)	20	0.48
11	75	F	Lt	21.1	House wife	−2.7	2	7	28.0	13,100 (69%)	12	0.11
12	76	F	Rt	29.8	House wife	−3.0	7	8	25.0	5810 (68%)	100	2.70
13	72	F	Rt	21.5	House wife	−2.6	2	8	25.0	5910 (71%)	26	0.01
14	81	M	Rt	23.2	Farmer	N	12	5	50.0	8890 (69%)	49	4.07
15	50	F	Lt	28.3	House wife	−5.2	12	6	39.2	6070 (62%)	75	1.60
16	80	F	Rt	25.0	House wife	−3.4	4	10	1.7	6290 (47%)	31	1.50
17	80	F	Rt	28.9	Farmer	−2.5	1	8	16.0	12,020 (80%)	9	0.03
18	73	F	Rt	25.4	House wife	−2.2	3	8	25.0	7570 (78%)	22	0.09
19	83	F	Rt	23.6	House wife	N	2	10	8.3	6480 (77%)	17	0.53
20	75	M	Rt	25.2	Farmer	N	5	5	39.2	8040 (77%)	54	4.00
21	69	F	Rt	26.4	Garment cutter	−2.9	2	5	26.7	4310 (74%)	26	0.01
22	75	F	Rt	22.6	Farmer	−5.8	12	9	23.3	5950 (60%)	16	0.01

F, female; M, male; Lt, left; Rt, right; BMI, body mass index; BMD, bone mineral density; Sx, symptom; VAS, visual analogue scale; ASES, American Shoulder and Elbow Surgeons; WBC, white blood cell; ESR, erythrocyte sedimentation rate; CRP, C-reactive protein.

**Table 2 diagnostics-10-00885-t002:** Radiographic findings based on plain radiographs, computed tomography, and magnetic resonance image.

Case	Subchondral Fracture (Low Signal Band)	Head Involvement	Head Collapse	Bone Marrow Edema	Joint Effusion	Bone Debris and Calcification	Synovitis	Rotator Cuff Tear	GHJ Arthritic Change	Glenoid Involvement
1	Y	Grade 3	Grade 1	Y	Y	Y	Y	SSP	N	N
2	NA	Grade 3	Grade 1	NA	Y	Y	Y	SSP, ISP	Y	N
3	Y	Grade 2	Grade 1	Y	Y	N	Y	SSC	N	N
4	Y	Grade 3	Grade 2	Y	Y	Y	Y	SSP, ISP	N	N
5	Y	Grade 2	Grade 2	Y	Y	N	Y	N	N	N
6	Y	Grade 2	Grade 1	Y	Y	N	Y	SSC	Y	N
7	Y	Grade 2	Grade 1	Y	Y	Y	Y	SSP, ISP	N	N
8	Y	Grade 1	Grade 1	Y	Y	Y	Y	N	N	N
9	Y	Grade 2	Grade 1	Y	Y	N	Y	SSP, ISP	N	N
10	Y	Grade 2	Grade 1	Y	Y	N	Y	SSP, ISP	N	N
11	Y	Grade 2	Grade 1	Y	Y	Y	Y	SSP, ISP, SSC	N	N
12	Y	Grade 3	Grade 2	Y	Y	Y	Y	SSP, ISP	Y	N
13	Y	Grade 2	Grade 1	Y	Y	Y	Y	SSP, ISP	N	N
14	Y	Grade 3	Grade 3	Y	Y	Y	Y	SSP, ISP	N	N
15	Y	Grade 3	Grade 1	Y	Y	N	Y	N	N	N
16	Y	Grade 3	Grade 2	Y	Y	N	Y	SSP	Y	N
17	Y	Grade 3	Grade 3	Y	Y	Y	Y	SSP	N	N
18	Y	Grade 3	Grade 3	Y	Y	Y	Y	N	N	N
19	Y	Grade 3	Grade 3	Y	Y	Y	Y	SSP, ISP	N	N
20	Y	Grade 3	Grade 3	Y	Y	N	Y	SSC, SSP	N	Y
21	Y	Grade 3	Grade 3	Y	Y	Y	Y	SSP, ISP	N	Y
22	Y	Grade 3	Grade 3	Y	Y	Y	Y	N	Y	Y

Y, yes; NA, not applicable; N, no; SSP, supraspinatus; ISP, infraspinatus; SSC, subscapularis; N, none; GHJ, glenohumeral joint.

**Table 3 diagnostics-10-00885-t003:** A review of literature.

Author	Case No.	Age	Sex	BMD	Time from Sx Onset to Head Collapse (month)	Bone Marrow Edema	Joint Effusion	Rotator Cuff Tear	Glenoid Involvement	Treatment
Tokuya et al. [2]	1	77	F	Osteoporosis	1	NR	Y	Y	Y	TSA
Yoshikawa et al. [5]	2	74	F	Osteoporosis	8	NR	Y	N	N	HA
		78	F	Osteoporosis	3	NR	Y	N	N	TSA
Goshima et al. [1]	2	77	F	Osteoporosis	5	Y	Y	Y	Y	HA
		74	F	Osteoporosis	5	Y	Y	N	N	TSA
Kakutani et al. [4]	2	81	F	Osteoporosis	1.5	NR	Y	N	N	HA
		81	F	Osteoporosis	1.5	NR	NR	NR	NR	HA
Kekatpure et al. [10]	9	72	F (all)	NR	5.7 (2–11)	Y (all)	Y (all)	Y (7)	N (all)	TSA (all)
		(63–85)						N (2)		
Kim et al. [11]	9	72.7	F (all)	Osteoporosis (7)	4.1 (1.2–5.9)	Y (all)	Y (all)	Y (6)	N (7)	RTSA (5)
		(57–78)		Osteopenia (2)				N (3)	Y (2)	TSA (4)
Our study	22	73.7	F (20)	Osteoporosis (14)	6.8 (1–12)	Y (all)	Y (all)	Y (17)	N (19)	RTSA (17)
		(50–83)	M (2)	Osteopenia (5)				N (5)	Y (3)	TSA (1)
				No BMD test (3)						HA (1)
										AS debridement (1)
										Conservative Tx (2)

F, female; M, male; BMD, bone mineral density; Sx, symptom; NR, not reported; Y, yes; N, no; TSA, total shoulder arthroplasty; HA, hemiarthroplasty; RTSA, reverse total shoulder arthroplasty; AS, arthroscopic; Tx, treatment.

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
