# Peer review of "Rapid Destructive Arthrosis Due to Subchondral Insufficiency Fracture of the Shoulder: Clinical Characteristics, Radiographic Appearances, and Outcomes of Treatment"

_diagnostics, 2020, doi:10.3390/diagnostics10110885_

Round 1

Reviewer 1 Report

Comments

The authors identified 22 patients (from 311 patient cohort) with RDA (criteria- rapid destruction was defined as > 25% erosive osteolysis of the humeral head 66 within 6 months according to Lequesne et al. or > 2 mm (or 50%) joint space narrowing in 12 months according to Postel and Kerboull) and SIF to characterize the patient characteristics and radiologic evidence to identify patient’s with RDA developed due to SIF.

Minor comments:

  1. How many patients were eliminated due to unavailability of data as opposed to not meeting the clinical requirement?

Major comments:

  1. Of 311 patients, the authors identified 22 that meet the clinical criteria. Of these 20 were female and 2 were male. The authors should clearly justify why they believe females are more prone to this?

  1. Instead of just presenting the data in tabular format, graphical representation might be more helpful

  1. Based on the authors aim, I do not see any specific patient characteristics shown to correlate with RDA due to SIF or what radiological evidences surgeons should look for to correctly diagnose this?

  1. Unclear aim and conclusions. The authors should clearly state the conclusions of the study from the data analyzed and results.

Author Response

Minor comments:

1.How many patients were eliminated due to unavailability of data as opposed to not meeting the clinical requirement?

Response: Thank you for the comment. This is a retrospective study included 311 patients who underwent or scheduled shoulder arthroplasty (including reverse total shoulder arthroplasty, anatomical total shoulder arthroplasty, and hemiarthroplasty) 2012 and 2019. Most patients had available data such as medical records and radiographic findings because all patients underwent or scheduled shoulder arthroplasty. When we found any destruction of humeral head, for diagnosis of RDA due to SIF, we thoroughly reviewed medical records and serial radiographic findings to rule out the diseases that can cause rapid destructive arthrosis and need to differentiate including inflammatory arthritis, osteonecrosis, infectious arthritis, crystal-induced destructive arthropathy. We added these sentences “Among of 311 patients, 289 patients were excluded including 215 cuff tear arthropathy or massive rotator cuff tear, 50 osteoarthritis, 16 infectious arthritis, and 8 osteonecrosis of the humeral head. Finally, 22 patients were included in this study.”

Major comments:

1.Of 311 patients, the authors identified 22 that meet the clinical criteria. Of these 20 were female and 2 were male. The authors should clearly justify why they believe females are more prone to this?

Response: Thank you for the comment. Previous studies demonstrated that SIF usually occurs in osteoporotic bone. In a review of the literature, most RDAs due to SIF of humeral head as well as femoral head occurred in women older than 70 years with bone fragility. As you know, osteoporosis predominantly occurs in women rather than men. That is why women are more prone. We described this point in Discussion.

2.Instead of just presenting the data in tabular format, graphical representation might be more helpful

Response: Thank you for the comment. According to your suggestion, we added Figure 2 as graph for incidences of radiographic findings.

3.Based on the authors aim, I do not see any specific patient characteristics shown to correlate with RDA due to SIF or what radiological evidences surgeons should look for to correctly diagnose this?

Response: Thank you for the comment. As you know, RDA due to SIF is a recently proposed disease and difficult to diagnose because of its rarity. In this study, we found these characteristics based on the medical records and radiographic findings.

  1. RDA due to SIF presented with severe short-term pain and functional disability.
  2. It commonly occurred in elderly women with bone fragility.
  3. MRI revealed bone marrow edema, extensive joint effusion, and synovitis as well as a diversity of types of head destruction with subchondral fracture within several months from onset of symptoms. (We also provided graphic figure (Figure 2) for incidence of radiologic findings.)

à These are typical findings of RDA due to SIF and different points with other diseases to need differential diagnosis.

4.Unclear aim and conclusions. The authors should clearly state the conclusions of the study from the data analyzed and results.

Response: Thank you for the comment. The primary aim of this study was to investigate the clinical characteristics and radiographic appearance in patients with RDA due to SIF of the shoulder. Based on the results, we restated the conclusion to “RDA due to SIF of the shoulder, presenting with severe short-term pain and functional disability, commonly occurred in elderly women with bone fragility. MRI revealed bone marrow edema, extensive joint effusion, and synovitis as well as a diversity of types of head destruction with subchondral fracture within several months from onset of symptoms.” We deleted the sentence in the Conclusion. “The results presented here indicate that SIF should be included in the differential diagnosis of acute onset shoulder pain in elderly patients.”

Reviewer 2 Report

Authors of the present study report aretrospective study evaluating RDA associated to SIF. This is a well written study.

More data are necessary concerning the retrospective institutional database review (how many patients were evaluated in total, how many were excluded and for what reasons etc)

Why IRB approval was necessary for this retrospective evaluation?

Authors claim that all available demographic and clinical evaluations were assessed however there is no mention of demographics' correlation as a potential risk factor or of any other association to RDA

Author Response

More data are necessary concerning the retrospective institutional database review (how many patients were evaluated in total, how many were excluded and for what reasons etc)

Response: Response: Thank you for the comment. This is a retrospective study included 311 patients who underwent or scheduled shoulder arthroplasty (including reverse total shoulder arthroplasty, anatomical total shoulder arthroplasty, and hemiarthroplasty) 2012 and 2019. Most patients had available data such as medical records and radiographic findings because all patients underwent or scheduled shoulder arthroplasty. When we found any destruction of humeral head, for diagnosis of RDA due to SIF, we thoroughly reviewed medical records and serial radiographic findings to rule out the diseases that can cause rapid destructive arthrosis and need to differentiate including inflammatory arthritis, osteonecrosis, infectious arthritis, crystal-induced destructive arthropathy. We added these sentences “Among of 311 patients, 289 patients were excluded including 215 cuff tear arthropathy or massive rotator cuff tear, 50 osteoarthritis, 16 infectious arthritis, and 8 osteonecrosis of the humeral head. Finally, 22 patients were included in this study.”

Why IRB approval was necessary for this retrospective evaluation?

Response: Thank you for the comment. Most medical journals will require IRB approval with patient’s consent before submission. So we prepared the document of IRB approval.

Authors claim that all available demographic and clinical evaluations were assessed however there is no mention of demographics' correlation as a potential risk factor or of any other association to RDA

Response: Thank you for the comment. The primary aim of this study was to investigate the clinical characteristics and radiographic appearance in patients with RDA due to SIF of the shoulder. We could not analyze potential risk factors of these conditions because of small number of cases. Based on the results, however, we could state the conclusion to “RDA due to SIF of the shoulder, presenting with severe short-term pain and functional disability, commonly occurred in elderly women with bone fragility. MRI revealed bone marrow edema, extensive joint effusion, and synovitis as well as a diversity of types of head destruction with subchondral fracture within several months from onset of symptoms.”
